# Alterations in the Cellular Metabolic Footprint Induced by Mayaro Virus

Ceyla M. O. Castro [1], Mânlio T. O. Mota [1], Alessandra Vidotto [2], Ícaro P. Caruso [3,4], Milene R. Ribeiro [1], Fábio R. Moraes [3], Fátima P. Souza [3] and Mauricio L. Nogueira [2,*]

[1] Virology Research Laboratory, School of Medicine of São José do Rio Preto (FAMERP), São José do Rio Preto 15090-000, SP, Brazil
[2] Multiuser Laboratory, School of Medicine of São José do Rio Preto (FAMERP), São José do Rio Preto 15090-000, SP, Brazil
[3] Multiuser Center for Biomolecular Innovation, Department of Physics, São Paulo State University (UNESP), São José do Rio Preto 15054-000, SP, Brazil
[4] Institute of Medical Biochemistry Leopoldo de Meis and National Center for Structural Biology and Bioimaging, Federal University of Rio de Janeiro, Rio de Janeiro 21.941-902, RJ, Brazil
* Correspondence: mauricio.nogueira@edu.famerp.br; Tel.: +55-(17)-3201-5731

**Abstract:** Mayaro virus is a neglected virus that causes a mild, dengue-like febrile syndrome characterized by fever, headache, rash, retro-orbital pain, vomiting, diarrhea, articular edemas, myalgia, and severe arthralgia, symptoms which may persist for months and become very debilitating. Though the virus is limited to forest areas and is most frequently transmitted by *Haemagogus* mosquitoes, *Aedes* mosquitoes can also transmit this virus and, therefore, it has the potential to spread to urban areas. This study focuses on the metabolic foot-printing of Vero cells infected with the Mayaro virus. Nuclear magnetic resonance combined with multivariate analytical methods and pattern recognition tools found that metabolic changes can be attributed to the effects of Mayaro virus infection on cell culture. The results suggest that several metabolite levels vary in infection conditions at different time points. There were important differences between the metabolic profile of non-infected and Mayaro-infected cells. These organic compounds are metabolites involved in the glycolysis pathway, the tricarboxylic acid cycle, the pentose phosphate pathway, and the oxidation pathway of fatty acids (via β-oxidation). This exometabolomic study has generated a biochemical profile reflecting the progressive cytopathological metabolic alterations induced by Mayaro virus replication in the cells and can contribute to the knowledge of the molecular mechanisms involved in viral pathogenesis.

**Keywords:** Mayaro virus; dengue-like fever; arboviruses; arthritogenic alphavirus; metabolomics; nuclear magnetic resonance

## 1. Introduction

Mayaro virus (MAYV) is a neglected arbovirus (ARthropod-BOrne virus) classified in the family *Togaviridae*, genus *Alphavirus*. This arthritogenic alphavirus causes a dengue-like febrile syndrome, sharing many symptoms with Chikungunya fever, including headache, rash, retro-orbital pain, vomiting, diarrhea, articular edemas usually associated with myalgia, and severe arthralgia/arthritis that can be very debilitating and can endure for months [1]. The viral genome is composed of a single-stranded, positive-sense RNA molecule. The virus has a cell-derived lipid enveloped around an icosahedral capsid that is 60–70 nm in diameter [2].

Since the first identification of the virus in Trinidad in 1954 [3], epidemiological and serological evidence of virus circulation has been found in Brazil (mainly in the Amazon region and the country's Central plateau) [4–8], Bolivia, Peru, Ecuador, Colombia, Venezuela, Trinidad, Guyana, French Guiana, Suriname, Panamá, Costa Rica, Honduras,

Guatemala, and Mexico [9–13]. It is thought that approximately 1% of all dengue-like illnesses in northern Latin America are caused by MAYV infections [14].

Despite its broad geographic dispersion, MAYV remains neglected in research and public health because of inadequate clinical and laboratory surveillance in endemic areas and the generic nature of its clinical manifestations [14,15]. These factors can cause Mayaro fever to be misdiagnosed as Dengue virus or another arbovirus [9,10].

The virus is maintained in nature in an enzootic cycle restricted to forest areas, and human infections are accidental. The major vectors are mosquitoes of the genus *Haemagogus*. However, MAYV has the potential for urbanization, since it can also be transmitted by the highly anthropophilic *Aedes* mosquitoes; there is therefore a risk of an explosive MAYV outbreak [10,16,17]. Furthermore, climate change (which affects the ecological niches of vectors), natural reservoirs, increased trade, and environmental degradation produced by deforestation may all contribute to the emergence of the MAYV in other areas [1,11,18,19]. MAYV typically causes small, sporadic outbreaks, but outbreak size is often more substantial in large cities with transmission cycles in urban and semi-urban areas [1,10,18,20,21].

Outbreaks had been reported throughout Latin America and Caribbean, with Brazil having most of the cases [22,23]. An outbreak of MAYV occurred in Manaus, a large city in the Amazon region, with nocturnal transmission cycles in urban and semi-urban areas [18]. This phenomenon was observed in the case of another closely related *Alphavirus*, CHIKV. This virus has adapted to mosquitoes of the genus *Aedes*, leading to the emergence of this virus in the Western Hemisphere and its rapid spread in South America and the Caribbean [24]. Cases of MAYV in patients without any contact with rural or sylvatic areas were reported in Mato Grosso [25]. In Cuiabá (a city in Mato Grosso) MAYV was isolated from *Ae. aegypti* mosquitoes. These viruses share 99–100% of their identity with sequences from humans from the same area [26]. The emergence of other important arboviruses as Zika virus (ZIKV) and Chikungunya virus (CHIKV) prompted more studies on arboviruses. These studies showed the number of MAYV cases is underestimated and is increasing in South America [22,23], especially in Brazil [27–31].

Furthermore, an autochthone MAYV case was reported in Haiti [32], an important route of the entrance of arbovirus in the southern USA as previously observed for CHIKV and ZIKV [6] raising concerns about a broader circulation of this virus to other countries in Central America and the Caribbean. It is important to highlight imported cases of MAYV were already reported in USA, Netherlands, Germany, France, and Switzerland confirming the global spread potential of MAYV [1].

Despite concern over the potential urbanization of MAYV, little is known regarding the basic aspects of its biology and replication. There are few studies on the molecular mechanisms involved in the pathophysiology of the arthralgia associated with the virus. This situation underscores the importance of a better knowledge of the mechanisms and metabolic pathways the virus requires for replication [1,33–36].

Despite the recent development of a rapid viral screening method [37], there are no specific treatments. There is a vaccine under development; however, it remains in the animal model stage [38]. The lack of an approved vaccine for Mayaro fever, combined with the potential urbanization of this virus indicates there is an urgent need for research into the virus's biology and replication.

Alterations in the extracellular medium are correlated with alterations in intracellular metabolism [39–41]. In this study, proton nuclear magnetic resonance (NMR) was used to find metabolic alterations in an extracellular media of cell cultures during MAYV replication.

This study used African green monkey (Vero) cells, a mammalian cell line in which the MAYV strain used in this study replicates well. The virus-induced metabolic alterations in this cell line are expected to be very similar to the alterations induced in humans. Three-time points were chosen to assess metabolic alterations at different stages of the viral replicative cycle: early (2 h), after one complete cycle (6 h), and late (12 h).

The results found provide information on which metabolic pathways the virus requires for replication and can shed light on these mechanisms, thus contributing to the development of new treatments or vaccines.

## 2. Materials and Methods

An outline of the experimental approach can be found in Figure 1.

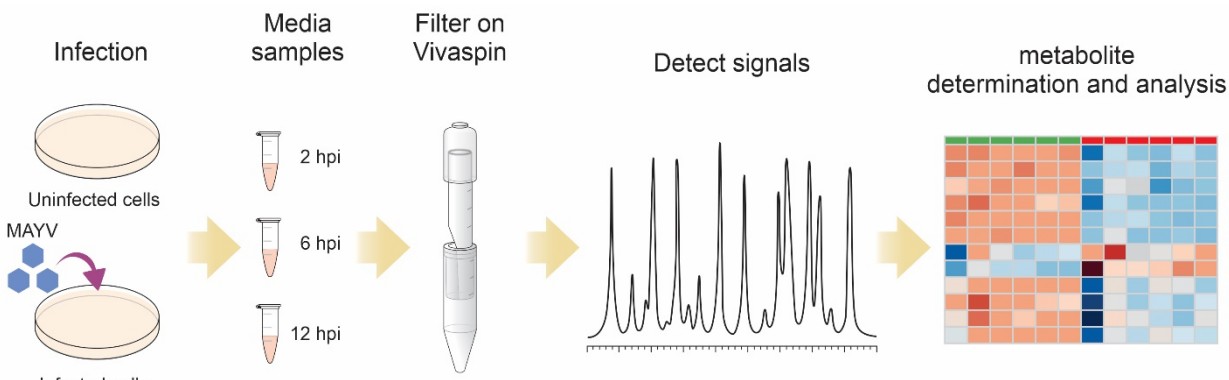

**Figure 1.** An outline of the approach used in this work. First Vero cell monolayers were infected with MAYV. In 2, 6, and 12 hpi samples were collected and filtered in Vivaspin. The filtrate was used to detect metabolites by NMR.

### 2.1. Cell Cultures and Virus

The viral strain BeAR-20.290 MAYV used in this study was isolated from Haemagogus mosquitoes in 1960 in Pará State, Brazil. It was initially propagated in suckling mice brains (Mus musculus), and then in C6/36 cells to produce the viral stock. This viral strain grows well in Vero cells, a mammalian cell line. It is expected that the metabolic alterations induced by the virus in this cell lineage be quite similar to that induced in humans [42].

Vero cells were seeded in six-well plates with MEM supplemented with 10% FBS for 24 h at 37 °C and 5% $CO_2$ with 95% confluence. Next, the monolayer was infected with MAYV at a multiplicity of infection (MOI) of 5 to guarantee that all cells in the monolayer would be infected. The Vero cell culture supernatant was collected at 2, 6, and 12 h post-infection (hpi). Vero E6 and C6/36 cell lines were both purchased from the American Type Culture Collection, or ATCC (Manassas, VA, USA). All the monolayers were treated using MEM and SBF from the same lot, and the cells were at the same passage for each time point collected. Six samples of normal cells and of infected cells were collected simultaneously for each time point.

### 2.2. Preparations of Samples for NMR

The Vivaspin filtration membrane has residual quantities of glycerine and sodium azide, which interfere with NMR analyses. Before use, the Vivaspin membranes were pre-washed 20 times with 2 mL of purified (deionized) water, and the tubes were stored in a refrigerator (4 °C) with purified water, covering the membrane surface until the NMR procedures.

All the infected and non-infected Vero cell supernatants collected at periods of 2, 6, and 12 hpi were thawed and added to the Vivaspin unit. They were then centrifuged at 4000 rpm for 10–15 min at 4 °C to filter extracellular metabolites. Finally, 550 μL of the filtrate was added to 50 μL of D2O and transferred to 5 mm tubes appropriate for NMR analyses.

### 2.3. Proton Nuclear Magnetic Resonance (NMR) Spectroscopy

NMR spectra were acquired in Bruker Avance HD III spectrometer operating at 600 MHz and equipped with a triple channel cryoprobe. The standard Bruker 1D pulse sequence NOESYPR1D was used with a mixing time of 100 ms. Sixteen scans were collected, as were four dummy scans. A spectral width of 14 ppm and 32k data points were used.

The relaxation delay was set to 5 s with 0.2 ms of gradient recovery. All measurements were performed at 293 K. Before Fourier transformation, a line broadening of 1 Hz was applied to each free induction decay (FID). Next, each spectrum was manually phase-corrected and referenced to the methyl doublet of lactate at 1.33 ppm.

To aid in metabolite identification, TOCSY spectra were performed using a DPSI sequence for mixing and pre-saturation for water suppression in selected samples. A mixing time of 80 ms was chosen, and spectra were collected using 64 scans with 4k data points in the direct dimension and 512 data points in the indirect dimension. Relaxation delay was maintained at 5 s. The $^{13}$C-$^{1}$H HSQC spectra also were acquired to support metabolite identification. Each spectrum was collected using 1k data points in the direct dimension and 256 in the indirect dimension, and 150 scans were recorded. A relaxation delay of 1.5 s was used, and a gradient recovery of 0.2 ms was applied. All spectra were processed in the TopSpin software, version 3.2 (Bruker, Germany).

### 2.4. Chemometric and Statistical Analyses

Binning was manually performed to remove noise and water signal regions and the ranges selected were 0.87–2.95 ppm, 3.00–4.01 ppm, 4.07–4.27 ppm, 4.57–4.60 ppm, 4.63–4.67 ppm, 5.22–5.28 ppm, 6.60–6.64 ppm, 6.87–6.91 ppm, 7.02–7.06 ppm, 7.16–7.22 ppm, 7.24–7.46 ppm, 7.52–7.57 ppm, 7.70–7.80 ppm, and 8.44–8.46 ppm.

The spectral ranges selected were uploaded onto the MetaboAnalyst web server [43,44] for further analysis. The dataset presented did not contain any missing data and underwent Pareto scaling in MetaboAnalyst. Principal component analysis (PCA) and partial least squares-discriminant analysis (PLS-DA) were used to first determine whether footprint-based metabolic differences were present in the control and infected cells and to then rank NMR signals for subsequent metabolite identification. PLS-DA is a supervised method and, as such, was validated using leave-one-out cross-validation. Both Welch's two-sample test ($p$-value $\leq 0.05$) and a fold change analysis (fold change less than 0.5 or higher than 2.0) were used to filter signals that underwent metabolite identification. Welch's two-sample test is performed by using the mean and standard error for each group as show in equation 1. The larger the $t$ value, the more likely it is for both groups to represent distinct groups, as regarding the used variable for calculation.

$$t = \frac{\overline{X}_1 - \overline{X}_2}{\sqrt{S_{X1}^2 + S_{X2}^2}} \tag{1}$$

where $\overline{X}_{1,2}$ is the meand and $S_{X_{1,2}}^2$ is the standard error.

Fold change is defined as simply the logarithm of the ratio between two averages, for example, between infected and control averages. The higher the fold change value, the more important it is for discriminating between groups. As well, for small values of fold change, it is understood that the corresponding signal has decreased in the infected group and, thus, it is potentially important.

Both indicators are combined into the so-called Volcano Plot, and, thus, signals that are potentially important for discrimination between control and infected are selected. It is important to note that both indicators ($p$-value and fold change) were used to select signals for metabolite identification, reducing the number of signals from in the thousands to twenty signals. No further conclusions regarding metabolic differences between conditions can be deduced from the indicators.

NMR signals passing the criteria listed above and ranked according to their PCA loadings were identified using the Chenomx NMR Suite, version 8.1 (Chenomx; Edmonton, AB, Canada). Each spectrum was manually visualized in the TopSpin software, version 3.2 (Bruker; Billerica, MA, USA), and no considerable resonance shifts across different samples were observed for the signals selected. After metabolite identification, specific resonances were selected to perform signal integrations. The signals selected were not in crowded

regions and could therefore be associated with each individual metabolite concentration in the cell culture medium.

Furthermore, to assess the consistency of the metabolites identified in the metabolic changes in cell culture following MAYV infection over time, a Support Vector Machine (SVM) algorithm for the classification of infected and control cell culture medium was used. The SVM algorithm, available from MetaboAnalyst, was applied and tested using Receiver Operating Characteristic (ROC) curves [45]. The Biomarker Analysis module in MetaboAnalyst was applied independently to the 2, 6, and 12 hpi datasets. The same scaling was used as previously mentioned. Combinations of metabolites were tested using the SVM approach and the MetaboAnalyst built-in feature selection. The ROC curve and 95% confidence interval were generated using a Monte Carlo cross-validation based on 70% of samples for feature selection and training and the remaining 30% of samples for testing, a process that was repeated multiple times.

## 3. Results

PCA and PLS-DA score plots with the first two principal components were successful in dividing the samples into two well-distinguished groups for all three-time points (2, 6, and 12 hpi), indicating that the NMR signals correspond to the metabolic state of the infection (Figure 2). This result also suggests that resonances from the $^1$H NMR spectrum (i.e., metabolite concentrations in cell culture medium) differ between infected and controlled cells at different time points.

Volcano plots of each time point contains both criteria used to select signals for metabolite identification, i.e., Welch's *p*-value of 5% or less and a fold change at a 2.0 threshold (Figure S1). The unique NMR signals that met both criteria of the Volcano plot increased to 100 at 2 hpi, 126 at 6 hpi, and 200 at 12 hpi.

Metabolite identification was performed using the Chenomx profiler software. The Chenomx library was searched to find signals that met the criteria of the volcano plot, and the findings were sorted by loadings. PLS-DA VIP scores were used to sort signals. To confirm whether the Chenomx results fit into the 1D NMR spectra, TOCSY spectra were used to check for cross-peaks. In addition, $^{13}$C-$^1$H-HSQC was used to assign metabolites to each specific signal when only one signal was present, as well as in crowded regions. The metabolites identified are summarized in Table 1. Figure 3 outlines reference spectra from infected cells 2, 6, and 12 hpi with identified metabolites (For a similar figure with control samples, please, refer to Figure S2 in Supplementary Material). Table 1 shows the relative concentration of the identified metabolites at each time point under study, and the corresponding boxplots are presented (Figures S3–S5). Following the assignment of NMR signals, the extra signals at 6 and 12 hpi that passed the criteria of the Volcano plot corresponded to very similar metabolites, as observed 2 hpi. Lactate was present only on the 6 hpi list. Isoleucine and leucine were found to be present at both 6 and 12 hpi, but not in samples after 2 hpi. In order to have relative levels of identified metabolites, the integration of the specific resonance signals for each metabolite was performed, and this result was used to compare metabolite levels between controls and infected cells (Figure 4). This second round of analyses was performed using only identified metabolites, reducing the number of signals to only twenty metabolites. Most metabolites were found to have similar relative levels between the controls and the infected cells. The levels of aspartate were not found to be affected by MAYV infection after 2 hpi, but they were affected at 12 hpi. Lactate levels presented changes only at 6 hpi, but not at the beginning or at the end of the infection process. In addition, PCA and PLS-DA loadings indicating important metabolites are shown in Figure S6. PLS-DA cross-validation results indicate that the metabolites identified were successful in correctly classifying controls and infected samples.

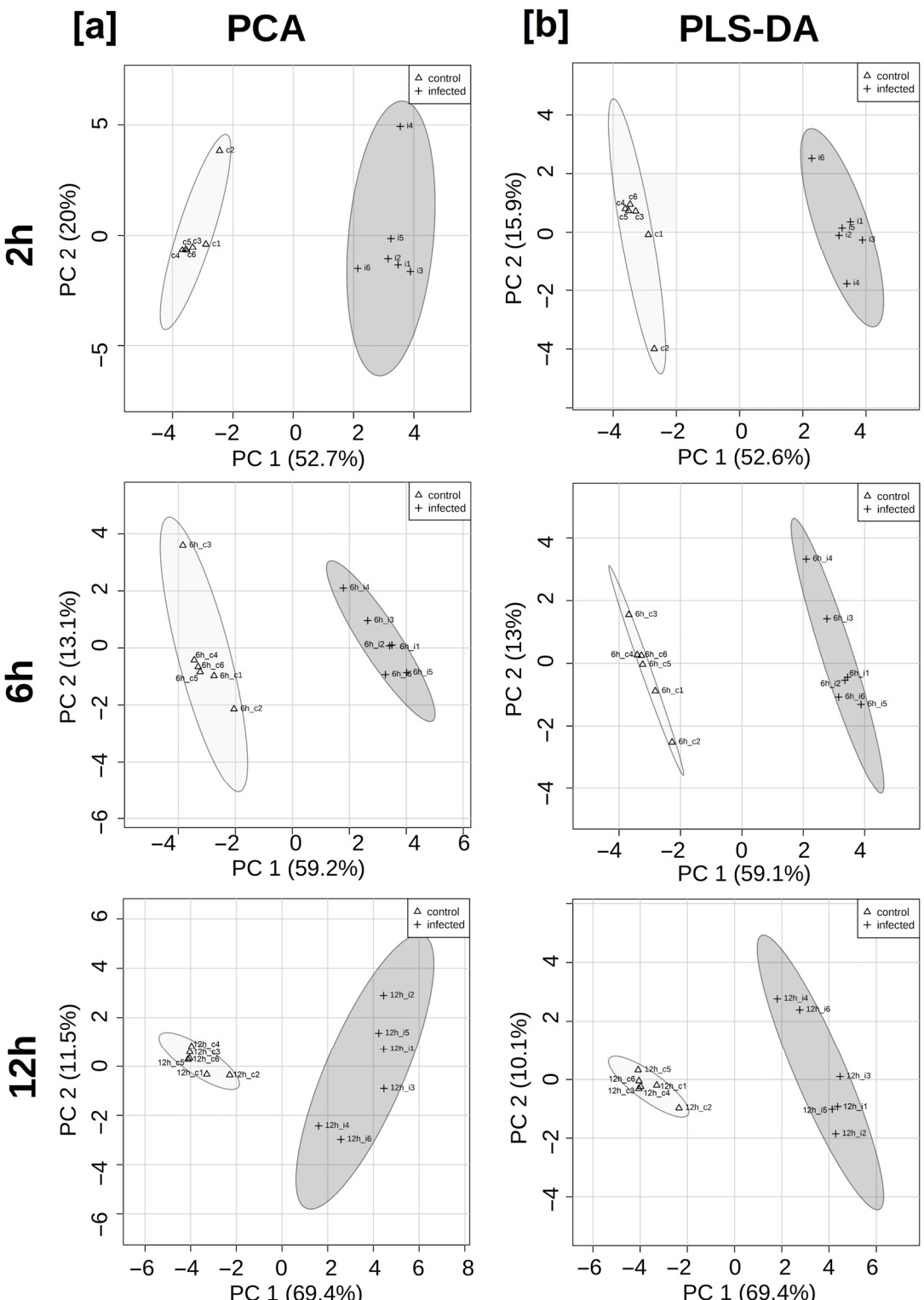

**Figure 2.** Summary of PCA and PLS-DA results. First and second principal components from PCA (**a**) and PLS-DA (**b**) after binning of the NMR spectra (all peaks, excluding water suppression region) of each time point: 2, 6, and 12 hpi. Accounted variability of each parameter is shown in brackets. Ellipses represent the 95% confidence interval for each group, control (triangle), and infected (plus) cells.

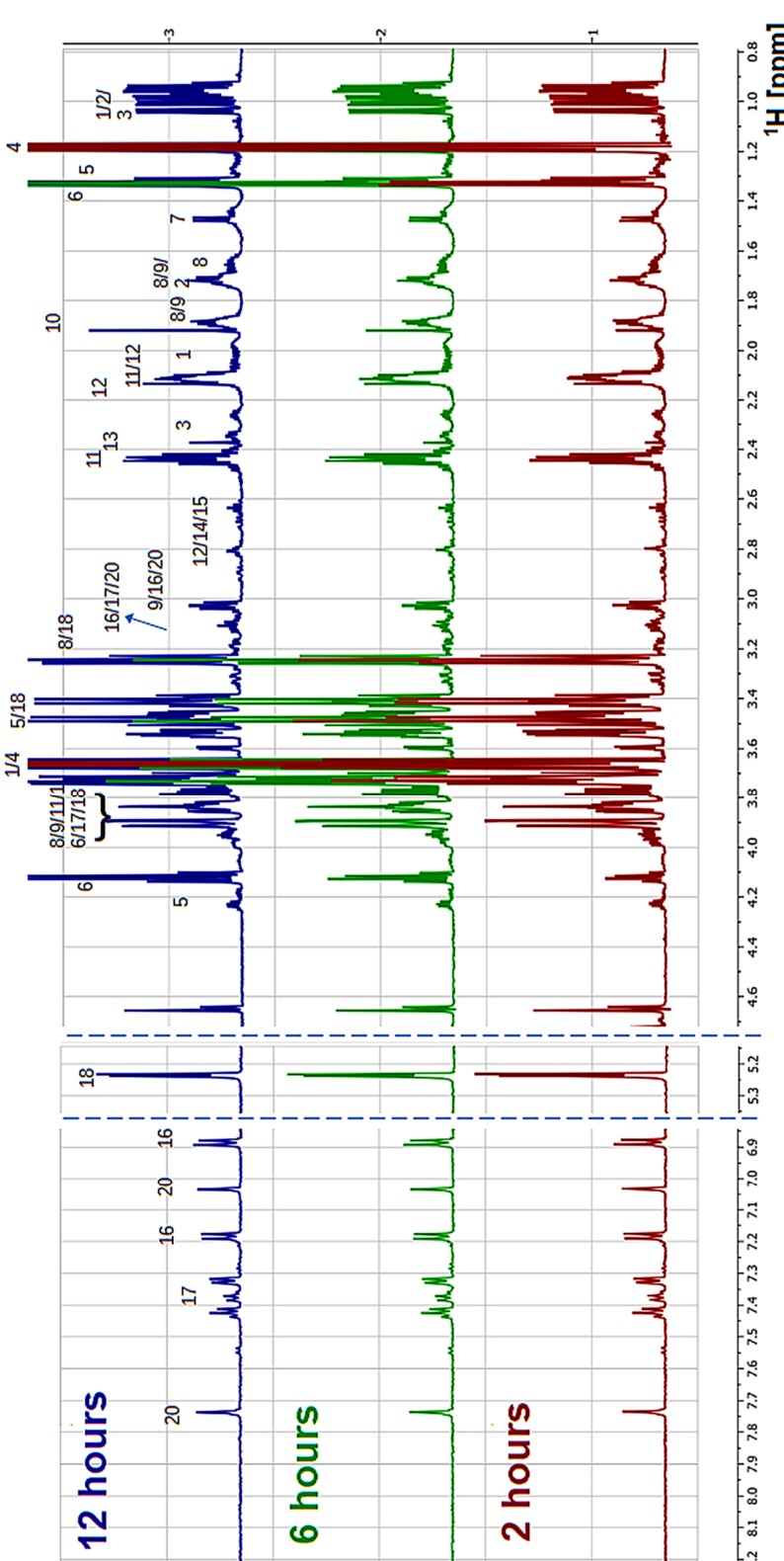

**Figure 3.** Reference NMR spectra from infected cells at 2 (brown), 6 (green), and 12 h (blue) post infection. Metabolite identification followed the aforementioned protocol. A total of 20 metabolites were identified as being relevant for the infection process: 1: Isoleucine, 2: Leucine, 3: Valine, 4: Ethanol, 5: Threonine, 6: Lactate, 7: Alanine, 8: Arginine, 9: Lysine, 1: Acetate, 11: Glutamine, 12: Methionine, 13: Pyruvate, 14: Aspartate, 15: Asparagine, 16: Tyrosine, 17: Phenylalanine, 18: Glucose, 19: Galactose, and 20: τ-Methylhistidine.

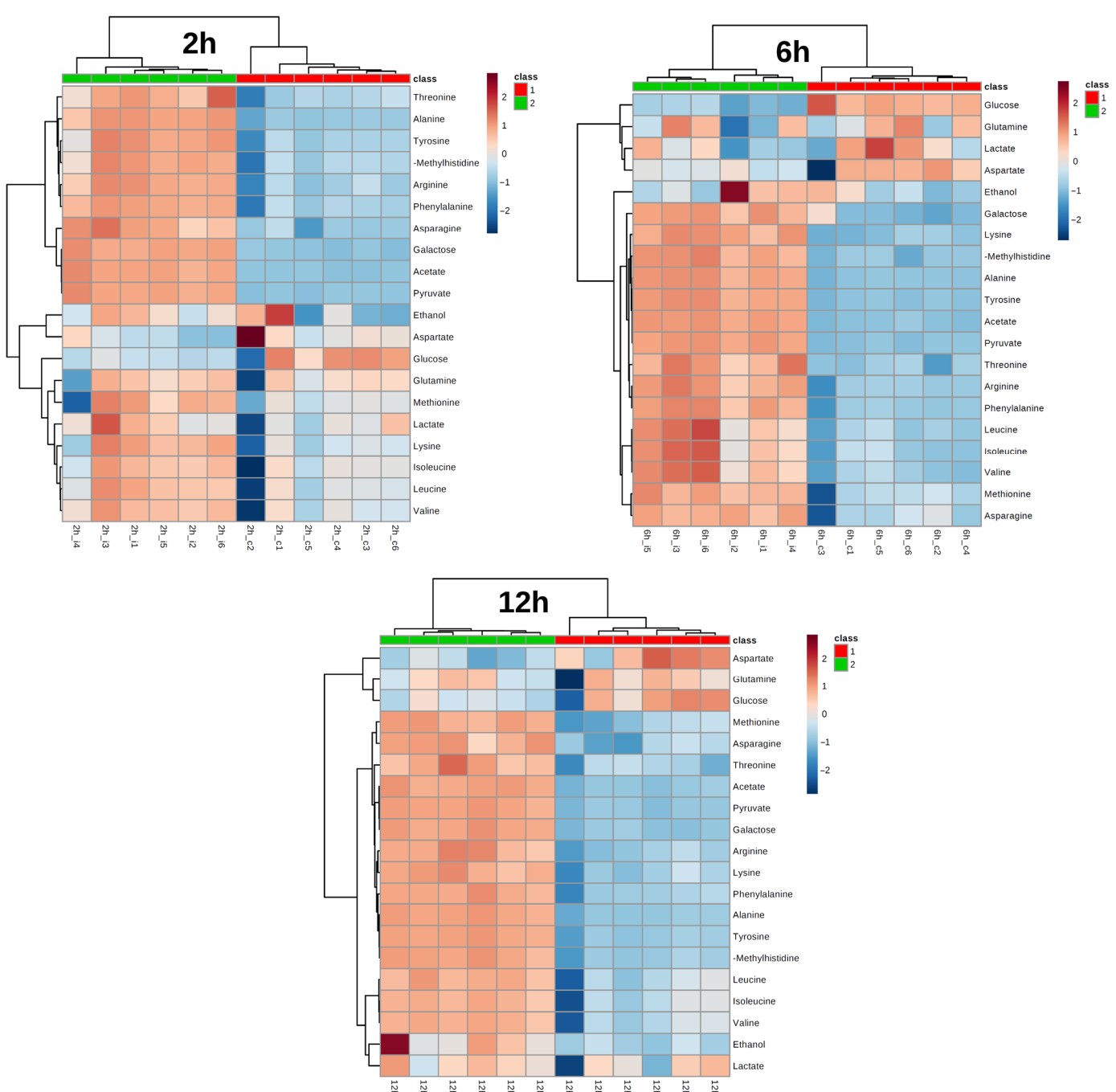

**Figure 4.** Levels of affected metabolite by MAYV infection process. The identified metabolites that are affected are evaluated by relative concentration for each specific time point. In 2 hpi (upper left) infected cells (label 2, green) are observed to have higher levels of most metabolites, except for aspartate, which level is not affected by infection, and glucose, which is more consumed in infected cells. In 6 hpi (upper right) infected cells are seen to consume more glucose (similarly to 2 hpi), lactate, and aspartate. Other identified metabolites are at higher levels in infected cells. In 12 hpi (bottom) infected cells follow the same behavior as in the 6 hpi time point, but levels are different between control and infected cells. Also, lactate is not observed to have different levels in control and infected cells.

The SVM algorithm was used to show that controls and infected cells at each time point have a different metabolic profile and was therefore successfully differentiated. This classification method is useful for highlighting how well the data used herein are

suitable for predicting new samples, as well as how robust the reported metabolites are for describing the infection process. Several individual metabolites were found to be able to correctly identify controls and infected samples at every post-infection time point studied (Figures S1–S5). To avoid overfitting issues, combinations of five metabolites were used to build the final SVM classifier in two distinct ways. In the first, MetaboAnalyst built-in SVM method was used to select metabolites. In the second, metabolites were manually chosen using a K-nearest neighbor algorithm to split them into five different groups to then select the highest $\log_2$ FC for each group. Figure 5 shows predicted class probabilities for each infection time using the SVM classifier with manually selected metabolites following the above-mentioned criteria. Results of the SVM algorithm with built-in feature selection are also available in the supplementary material (Figures S7–S9). The ROC curves for each classifier in Figure 5 indicate that the metabolic changes observed are consistent and were able to differentiate between controls and infected cells at every time point.

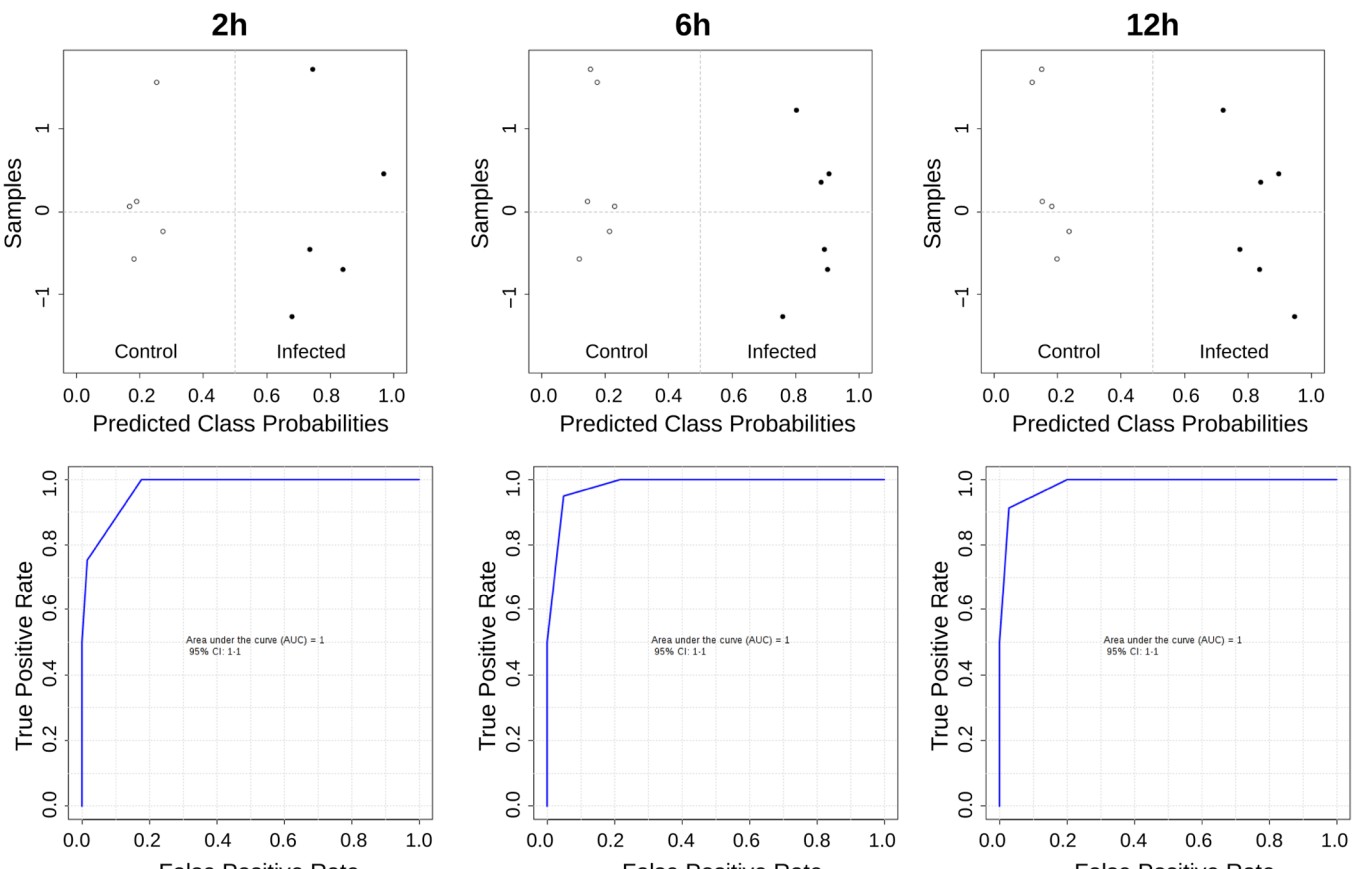

**Figure 5.** SVM classification error of MAYV infected cells. Assessment of model quality through classification using SVM. Classification error is related to the robustness of the metabolites identified in the chemometric analysis in predicting a metabolic profile in control or infected cell. The classification error reduces according to the progress of the infection process.

The metabolomic approach used in this study revealed a metabolic profile with cytopathological alterations caused by MAYV replication in Vero cells (Table 1). The metabolic footprints obtained are attributed to variations in metabolite levels detected and identified in infected Vero cell supernatants. This variation in the level of these organic compounds can be attributed to the effects of MAYV infection on Vero cells.

**Table 1.** Relative metabolite levels as integrals measured for specific resonances.

| Metabolite | 2 hpi | | 6 hpi | | 12 hpi | |
|---|---|---|---|---|---|---|
| | Control | Infected | Control | Infected | Control | Infected |
| Leucine | 9.48 ± 1.66 | 11.41 ± 0.71 | 10.64 ± 0.17 | 11.60 ± 0.40 | 6.45 ± 0.72 | 7.87 ± 0.15 |
| Isoleucine | 6.71 ± 1.30 | 7.93 ± 0.53 | 7.47 ± 0.15 | 8.11 ± 0.25 | 6.79 ± 0.82 | 8.21 ± 0.13 |
| Valine | 6.76 ± 1.17 | 8.12 ± 0.34 | 7.41 ± 0.15 | 8.23 ± 0.28 | 6.74 ± 0.84 | 8.38 ± 0.13 |
| Ethanol | 102.90 ± 23.81 | 110.75 ± 9.51 | 103.40 ± 18.91 | 123.10 ± 33.30 | 75.54 ± 9.87 | 129.52 ± 40.60 |
| Threonine | 3.78 ± 0.76 | 6.20 ± 0.69 | 4.31 ± 0.30 | 6.26 ± 0.41 | 4.78 ± 0.53 | 6.68 ± 0.38 |
| Lactate | 8.13 ± 1.07 | 9.08 ± 0.65 | 18.39 ± 1.02 | 17.74 ± 0.76 | 31.58 ± 4.43 | 34.15 ± 1.66 |
| Alanine | 3.48 ± 0.71 | 10.72 ± 0.74 | 4.82 ± 0.27 | 10.94 ± 0.48 | 4.81 ± 0.59 | 11.21 ± 0.33 |
| Arginine | 0.37 ± 0.09 | 0.70 ± 0.05 | 0.42 ± 0.06 | 0.73 ± 0.05 | 0.38 ± 0.06 | 0.71 ± 0.05 |
| Acetate | 1.42 ± 0.09 | 7.32 ± 0.41 | 2.67 ± 0.25 | 8.91 ± 0.29 | 4.06 ± 0.56 | 12.45 ± 0.46 |
| Pyruvate | 0.50 ± 0.10 | 6.01 ± 0.35 | 0.92 ± 0.10 | 4.09 ± 0.16 | 1.44 ± 0.10 | 3.22 ± 0.10 |
| Glutamine | 3.05 ± 0.80 | 3.35 ± 0.55 | 3.51 ± 0.10 | 3.47 ± 0.14 | 3.05 ± 0.34 | 3.09 ± 0.12 |
| Methionine | 0.97 ± 0.21 | 1.25 ± 0.55 | 0.93 ± 0.26 | 1.48 ± 0.07 | 0.94 ± 0.16 | 1.55 ± 0.05 |
| Aspartate | 0.34 ± 0.22 | 0.15 ± 0.10 | 0.15 ± 0.25 | 0.08 ± 0.03 | 0.22 ± 0.09 | 0.07 ± 0.04 |
| Asparagine | 0.35 ± 0.07 | 0.76 ± 0.08 | 0.22 ± 0.23 | 0.70 ± 0.05 | 0.28 ± 0.14 | 0.79 ± 0.08 |
| Lysine | 1.24 ± 0.30 | 1.70 ± 0.27 | 1.42 ± 0.04 | 1.78 ± 0.04 | 1.34 ± 0.10 | 1.73 ± 0.04 |
| Glucose | 10.13 ± 1.53 | 9.10 ± 0.22 | 10.20 ± 0.34 | 8.40 ± 0.32 | 7.70 ± 1.19 | 7.10 ± 0.28 |
| Galactose | 0.04 ± 0.03 | 1.10 ± 0.06 | 0.22 ± 0.23 | 1.00 ± 0.09 | 0.10 ± 0.05 | 0.93 ± 0.04 |
| Tyrosine | 3.02 ± 0.66 | 5.71 ± 0.72 | 3.42 ± 0.10 | 5.94 ± 0.19 | 3.17 ± 0.38 | 5.97 ± 0.12 |
| Phenylalanine | 2.68 ± 0.40 | 4.03 ± 0.10 | 2.92 ± 0.16 | 3.99 ± 0.16 | 2.68 ± 0.32 | 4.01 ± 0.11 |
| τ-Methylhistidine | 2.07 ± 0.50 | 3.56 ± 0.33 | 2.34 ± 0.13 | 3.69 ± 0.17 | 2.19 ± 0.26 | 3.81 ± 0.12 |

## 4. Discussion

The profiling of metabolite secretion reflects the cellular metabolic activity, providing insights into intracellular metabolic processes and variations in metabolite levels in infected cells relative to the control. These processes and variations represent the effect of cellular biochemical reactions (anabolism and catabolism) that occur in response to the virus infection, suggesting affected metabolic pathways and possible mechanisms of action of the virus replication [39–41]. At this point is impossible to determine the exact metabolite pathways influenced by MAYV replication; however, the results can throw some light on it. In this report, we observed variations in the levels of 20 metabolites in the three-time points studied of the MAYV infection.

We observed alterations in the amino acid metabolism, which is expected of any virus infection. The increase and decrease in amino acids are not specific to a metabolic pathway; however, it is possible to infer some possibilities. For example, an increase in glutamine was observed. This amino acid is the main biological source of amino groups for a wide array of biosynthetic processes and plays a central role in the metabolism of amino acids in mammals [46]. Therefore, increased glutamine affects the metabolism of amino acids, and this finding herein may be associated with the increases in amino acids observed during the three-time points.

Methionine, arginine, and lysine were detected at increased levels in infected samples at 2, 6, and 12 hpi. They participate in anabolic and catabolic TCA cycle pathways; methionine and arginine are glycogenic, and lysine is ketogenic [47]. The high levels of methionine and arginine found in the infected cell samples may be associated with the disruption of cell homeostasis, energy production, and the consumption mechanism (ATP) [48,49]. The increases in methionine, arginine, and lysine in the infected samples may be linked to glutamine levels since this is central in the metabolism of amino acids. The increase in arginine in the infected samples may be associated with acetate and acetic acid that was detected at highly differentiated levels.

Valine was also found to increase in infected samples when measured at different time points. Higher levels of valine and pyruvate may be associated with changes in glucose metabolism. Studies have demonstrated that MAYV alters glucose metabolism through the enzyme 6-phosphofructo 1-kinase [50].

Furthermore, the pattern of metabolic alterations presented in this study allowed us to infer that MAYV replication in Vero cells interferes with the glycolysis and TCA pathways, both of which are important for cellular energy and biosynthesis. It is known that other alphaviruses, such as the Semliki Forest virus and Sindbis virus, are also known to interfere with these pathways [51].

The data collected in this study is not sufficient to support the conclusion that MAYV interferes with lipogenesis. However, as alphavirus are enveloped viruses, it is expected that MAYV can cause alterations in this metabolic pathway since lipid and cholesterol metabolism are important for the entry and replication of enveloped viruses [52–55].

Acetate or acetic acid is a carboxylic acid that is involved in pyruvate metabolism and lipogenesis [56]. It was found to increase during the three infection periods. Acetyl-CoA hydrolysis produces the acetate, and acetyl-CoA is a key intermediary in biochemical reactions in the glycolytic pathway, the tricarboxylic acid (TCA) cycle, the β-oxidation pathway, and the lipogenesis [57–59]. This observation supports the hypothesis the high level of acetate at the three-time points may be associated with acetyl-CoA regulation and the virus replication process. However, more studies need to be carried out to confirm this hypothesis.

As also expected, enveloped virus interferes with the cytoskeleton organization and the integrity of the cytoplasm membrane. Decreases in glucose and increases in galactose were identified and detected in infected samples relative to the control samples. The release of viral particles takes place at 6 hpi, and there was an increase in the permeability of the plasmatic membrane at the time of the entrance of the virus, perhaps due to the formation of pores by the viral proteins E1 or 6K. The formation of syncytia ("polykaryocyte") by the Semliki Forest virus, an alphavirus was already observed [60].

The presence of syncytia and the acidification of the extracellular media were observed at 6 hpi (data not shown). These observations, along with the increase in galactose, support the hypothesis that these metabolites may be regulating ATP production, which interferes with syncytia formation.

Furthermore, these results are in accordance with the enzyme analyses, which revealed changes in the glucose metabolism of Vero cells infected with MAYV [51]. It is important to note that, when galactose is transformed into an intermediate glycolytic metabolite, it acts upon the metabolism of nucleotide sugars. This observation suggests that the later-stage effects of the infection may be influencing the biochemical regulation of nucleotide sugar metabolism. Another metabolite, τ-methylhistidine, a component of the actin and myosin filament, was altered in infected samples at all-time points.

NMR spectroscopy combined with multivariate analytical methods revealed important differences between infected and non-infected cells. These data showed that the exometabolome generates a biochemical profile that reflects the metabolic status of the infected cells at different time points. These alterations are linked to the progression of cytopathic effects of the virus on the cells. Our study is preliminary, and future research to determine any associations between genomic and proteomic data is necessary to clarify the molecular mechanisms involved in MAYV pathogenesis.

**Supplementary Materials:** The following supporting information can be downloaded at: https://www.mdpi.com/article/10.3390/biomed3010013/s1, Figure S1: Volcano plot and Venn Diagram for highlighting important NMR signals. Time-specific volcano plot (left) for selecting NMR signals that attend the criteria of *p*-value, from Welch-Two Sample test, less than 5% and a fold change threshold of 2. The Venn diagram (right) shows an increasing number of important signals that are observed according to the progress of the infection process. Nevertheless, only three new metabolites (lactate, isoleucine, and leucine) are seen in 6h p.i. and 12h p.i.. The lactate pathway was only affected in the 6h p.i. cells.; Figure S2. Reference NMR spectra from control subjects at 2 (brown), 6 (green), and 12 h (blue) post infection. Metabolite identification followed the protocol mentioned in the main text. A total of 20 metabolites were identified as being relevant for the infection process: 1: Isoleucine, 2: Leucine, 3: Valine, 4: Ethanol, 5: Threonine, 6: Lactate, 7: Alanine, 8: Arginine, 9: Lysine, 1: Acetate, 11: Glutamine, 12: Methionine, 13: Pyruvate, 14: Aspartate, 15: Asparagine, 16: Tyrosine, 17: Phenylalanine, 18: Glucose, 19: Galactose, and 20: τ-Methylhistidine; Figure S3: Boxplots indicating

metabolites concentrations at 2 hpi.; Figure S4: Boxplots indicating metabolites concentrations at 6 hpi.; Figure S5: Boxplots indicating metabolites concentrations at 12 hpi.; Figure S6: Loadings from PCA and PLS-DA for each time point. Metabolites with high values are Acetate, Glucose, Pyruvate, Glutamine, Ethanol, and Lactate for all time points. The bottom shows the PLS-DA performance in leave-one-out cross-validation for each time post-infection; Figure S7: SVM classification of MAYV infected cells at 2 hpi. Assessment of model quality through classification using SVM with its built-in variable selection approach and evaluation by means of area under the ROC curves. Classification performance is related to the robustness of the metabolites identified in the chemometric analysis in predicting a metabolic profile in control or infected cell; Figure S8: SVM classification of MAYV infected cells at 6 hpi. Assessment of model quality through classification using SVM with its built-in variable selection approach and evaluation by means of area under the ROC curves. Classification performance is related to the robustness of the metabolites identified in the chemometric analysis in predicting a metabolic profile in control or infected cell.; Figure S9: SVM classification of MAYV infected cells at 12 hpi. Assessment of model quality through classification using SVM with its built-in variable selection approach and evaluation by means of area under the ROC curves. Classification performance is related to the robustness of the metabolites identified in the chemometric analysis in predicting a metabolic profile in control or infected cell.

**Author Contributions:** Conceptualization, M.T.O.M., C.M.O.C., A.V., Í.P.C., M.R.R., F.R.M., F.P.S., and M.L.N.; methodology, M.T.O.M., C.M.O.C., Í.P.C., M.R.R., and F.R.M.; formal analysis, M.T.O.M., C.M.O.C., A.V., Í.P.C., M.R.R., F.R.M., F.P.S., and M.L.N.; investigation, M.T.O.M., C.M.O.C., Í.P.C., M.R.R., and F.R.M.; writing—original draft preparation, M.T.O.M. and C.M.O.C.; writing—review and editing, M.T.O.M., C.M.O.C., A.V., Í.P.C., M.R.R., F.R.M., F.P.S., and M.L.N.; funding acquisition, F.P.S. and M.L.N. All authors have read and agreed to the published version of the manuscript.

**Funding:** This work was supported by the São Paulo Research Foundation (FAPESP; grant numbers 2013/21793, 2014/05600-9, and 2009/53989-4), the Association for the Improvement of Higher Education Personnel (CAPES) scholarship awarded to C.M.O.C. and the Brazilian National Council for Scientific and Technological Development (CNPq) fellowship awarded to M.L.N.

**Institutional Review Board Statement:** Not applicable.

**Informed Consent Statement:** Not applicable.

**Data Availability Statement:** The data that support the findings of this study are available on the link https://www.ebi.ac.uk/metabolights/MTBLS639. Accessed on 19 December 2022.

**Conflicts of Interest:** The authors declare no conflict of interest.

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
