# Peer review of "Alterations in the Cellular Metabolic Footprint Induced by Mayaro Virus"

_2673-8430, doi:10.3390/biomed3010013_

Round 1

Reviewer 1 Report

I don't think the Figure 6 ASCA panels (their in-house method) is informative or answers the question they are asking, and I'm worried that time differences themselves could make their way into the components, leading the metod to simply ask whether the time points have any differences, which isn't what you want to know, you want to know if the same infected-vs-control differences remain, at one signal-to-noise ratio or another, at all time points. Also needs more methods in the legend or somewhere. However, this figure is not critical, as Figure 4 covers the same topic.

Vero cells don't produce interferon, how does this affect the results? Wouldn't interferon counteract the changes you see? Would you expect cells that managed an interferon response to be infected in the first place, so that the question is relevant?

Are the results the same as you'd get from infection with an unrelated virus? More glycolysis and basic biosynthetic pathways are kind of what you'd expect from a cell that starts producing virions, and I think HIV produces a similar effect (from skimming https://pubmed.ncbi.nlm.nih.gov/33117366/). The discussion section touches upon the needs of an enveloped virus, but more comparison would be nice.

Author Response

First, we would like to thank the reviewer for your considerations.

"I don't think the Figure 6 ASCA panels (their in-house method) is informative or answers the question they are asking, and I'm worried that time differences themselves could make their way into the components, leading the metod to simply ask whether the time points have any differences, which isn't what you want to know, you want to know if the same infected-vs-control differences remain, at one signal-to-noise ratio or another, at all time points. Also needs more methods in the legend or somewhere. However, this figure is not critical, as Figure 4 covers the same topic."

Answer: After reading the reviewer's considerations, we decided to remove the ASCA analysis from the new version of the manuscript.

"Vero cells don't produce interferon, how does this affect the results? Wouldn't interferon counteract the changes you see? Would you expect cells that managed an interferon response to be infected in the first place, so that the question is relevant?"

Answer: Despite the high probability of MAYV causing major outbreaks it remains neglected, there are few studies about it. The cell alterations induced by the MAYV are largely unknown. This study tries to get a glimpse of the cytopathic modifications or alterations induced by virus infection. For sure interferon plays an important role in virus replication, however, interferences and blockages induced by interferon only can be correctly accessed by comparison with basal values. This kind of comparison is the next step of the study.

"Are the results the same as you'd get from infection with an unrelated virus? More glycolysis and basic biosynthetic pathways are kind of what you'd expect from a cell that starts producing virions, and I think HIV produces a similar effect (from skimming https://pubmed.ncbi.nlm.nih.gov/33117366/). The discussion section touches upon the needs of an enveloped virus, but more comparison would be nice."

Answer: Again, the lack of knowledge about the virus-induced alterations plays against us. Due to the redundancy of the cell metabolism, it is impossible, at the stage of this work, to point to a specific metabolic pathway as responsible for the alterations in the metabolite levels observed. Comparisons between MAYV, HIV, and any other unrelated virus that may share the same characteristics would be just a guess and will not provide useful information.

Reviewer 2 Report

In this paper the authors described a metabolomic analysis of the growth media of  Vero cells infected with the Mayaro virus, by Nuclear magnetic resonance combined with multivariate analysis. They monitored the metabolic alterations at different stages of the viral cycle: early (2 hours), after one complete cycle (6 hours), and late (12 hours).

Although they have obtained some interesting data, the presentation and discussion of the results are confused and not well organized.

First of all they must describe the NMR spectra in the three stages: they reported NMR spectra only in Figure 3 for 12 hpi with identified metabolites (and this information is not reported in the caption), but they must add the spectra of the other two stages at least in the supporting information.

After the presentation of the NMR results, the PCA and PLS-DA score plots (Figure2) can be introduced and discussed. The anlaysis of the Volcano plots reported in the SI must be well described in the text.

The ASCA analysis must be well described in the text in order to highlight the difference in concentration of the metabolites in the same cell line over the time and between different cell lines (control and infected).

My main concern is the discussion of the results: they commented the correlation of the observed metabolites variations with the glycolysis and TCA pathways, the lipogenesis, the pyruvate metabolism, but the data collected here are very generic (the variation include all the amino acids, the lactate and the piruvate) and are not sufficient to support the conclusions. 

I agree with the latest paragraph of the discussion "NMR spectroscopy combined with multivariate analytical methods revealed important differences between infected and non-infected cells", which must be moved at the beginning of this paragraph since it is the only clear conclusion that they can extrapolate.

In the discussion they must underline the meaning of the observed metabolic differences in the cycle of the infected cells and those observed with respect to the control cells.

Author Response

"In this paper the authors described a metabolomic analysis of the growth media of  Vero cells infected with the Mayaro virus, by Nuclear magnetic resonance combined with multivariate analysis. They monitored the metabolic alterations at different stages of the viral cycle: early (2 hours), after one complete cycle (6 hours), and late (12 hours).

Although they have obtained some interesting data, the presentation and discussion of the results are confused and not well organized."

Answer: First, we would like to thank the reviewer for your considerations.

"First of all they must describe the NMR spectra in the three stages: they reported NMR spectra only in Figure 3 for 12 hpi with identified metabolites (and this information is not reported in the caption), but they must add the spectra of the other two stages at least in the supporting information."

Answer: This change was performed as suggested by the reviewer.

"After the presentation of the NMR results, the PCA and PLS-DA score plots (Figure2) can be introduced and discussed. The anlaysis of the Volcano plots reported in the SI must be well described in the text."

Answer: These changes were performed as suggested by the reviewer.

"The ASCA analysis must be well described in the text in order to highlight the difference in concentration of the metabolites in the same cell line over the time and between different cell lines (control and infected)."

Answer: As the other reviewer also criticized the ASCA analysis and suggested that its results were not necessary for the final conclusion, we decided to remove all the parts related to this analysis in the new version of the manuscript.

"My main concern is the discussion of the results: they commented the correlation of the observed metabolites variations with the glycolysis and TCA pathways, the lipogenesis, the pyruvate metabolism, but the data collected here are very generic (the variation include all the amino acids, the lactate and the piruvate) and are not sufficient to support the conclusions."

Answer: It's true that the correlation made between metabolite variation and some metabolic pathways is not fully supported by the observations. Due to the initial state of this work, this correlation seems to be the better one that explains the variations. We have made changes to the text of the new version of the manuscript to make it clear that the correlation is an assumption. The best that explains it, but an assumption.

"I agree with the latest paragraph of the discussion "NMR spectroscopy combined with multivariate analytical methods revealed important differences between infected and non-infected cells", which must be moved at the beginning of this paragraph since it is the only clear conclusion that they can extrapolate."

Answer: The phrase was moved to the beginning of the paragraph.

"In the discussion they must underline the meaning of the observed metabolic differences in the cycle of the infected cells and those observed with respect to the control cells."

Answer: The cell metabolic mechanism is redundant. At this point, it’s impossible to define, for each metabolite, which specific pathway was affected by virus infection. So, only a broad meaning for each alteration observed can be inferred and possible associations between metabolic alterations with the infectious process are suggested. A precise meaning relies on the identification of the metabolic pathway affected. Describing every possible meaning of every alteration would make the discussion text long, confusing, and boring and will not generate any valid information.

Round 2

Reviewer 2 Report

The authors have addressed all my comments and the paper can be published